# Dietary Patterns and Metabolic Disorders in Polish Adults with Multiple Sclerosis

**DOI:** 10.3390/nu14091927

**Published:** 2022-05-04

**Authors:** Edyta Suliga, Waldemar Brola, Kamila Sobaś, Elżbieta Cieśla, Elżbieta Jasińska, Katarzyna Gołuch, Stanisław Głuszek

**Affiliations:** 1Department of Nutrition and Dietetics, Collegium Medicum, Jan Kochanowski University, ul, Żeromskiego 5, 25-369 Kielce, Poland; waldemar.brola@ujk.edu.pl (W.B.); kamila.sobas@ujk.edu.pl (K.S.); eciesla@ujk.edu.pl (E.C.); elzbieta.jasinska@ujk.edu.pl (E.J.); 2RESMEDICA, 25-726 Kielce, Poland; kgoluch@interia.pl; 3Laboratory of Medical Genetics, Department of Surgical Medicine, Collegium Medicum, Jan Kochanowski University, ul, Żeromskiego 5, 25-369 Kielce, Poland; stanislaw.gluszek@ujk.edu.pl

**Keywords:** dietary patterns, multiple sclerosis, adults, metabolic risk factors

## Abstract

Diet plays a major role in the aetiopathogenesis of many neurological diseases and may exacerbate their symptoms by inducing the occurrence of metabolic disorders. The results of research on the role of diet in the course of multiple sclerosis (MS) are ambiguous, and there is still no consensus concerning dietary recommendations for patients with MS. The aim of this study was to analyse the dietary patterns (DPs) of patients with MS and to assess the relationships between these DPs and the metabolic disorders. The study participants were comprised of 330 patients aged 41.9 ± 10.8 years. A survey questionnaire was used to collect data related to diet, lifestyle and health. The DPs were identified using a principal component analysis (PCA). Three DPs were identified: Traditional Polish, Prudent and Fast Food & Convenience Food. An analysis of the odds ratios adjusted for age, gender, smoking and education showed that a patient’s adherence to the Traditional Polish and the Fast Food & Convenience Food DPs increased the likelihood of abdominal obesity and low HDL-cholesterol concentration. Conversely, adherence to the Prudent DP was not significantly associated with any metabolic disorder. The results of this study confirmed that an unhealthy diet in patients with MS is connected with the presence of some metabolic risk factors. There is also an urgent need to educate patients with MS on healthy eating, because the appropriate modifications to their diet may improve their metabolic profile and clinical outcomes.

## 1. Introduction

Multiple sclerosis (MS) is a progressive inflammatory-demyelinating disease of the central nervous system with an unknown aetiology. A characteristic feature of the disease is the presence of diffuse foci of demyelination, located primarily in the white matter of the brain, most often periventricularly [1,2]. In the initial stage of the disease, about 85% of cases are the relapsing-remitting form (RRMS), which after a certain time (that varies depending on the patient) develops into the secondary progressive form (SPMS) with a constant progression of disability [3,4,5]. The primary progressive form (PPMS) is characterised by a systematic build-up of disability from the onset of the disease, usually without overlapping relapses. It is a rarer form generally diagnosed much later than RRMS, with a characteristic symptomatology and a different image in magnetic resonance imaging (MDI) [6,7]. The dominant neurological symptoms are a growing spastic paresis of the lower limbs, and changes in MRI mainly related to the spinal cord. It is estimated that PPMS constitutes 10–20% of all cases of MS [6,7,8]. The onset of MS usually occurs between the ages of 20 and 40 years, and the disease is one of the most common causes of neurological disability in young individuals. The number of patients with MS exceeds 2.8 million globally, of whom about 1.0 million live in Europe. Recent epidemiological research has indicated that the prevalence of MS in Poland is over 120 cases per 100,000 people, for a total of about 45,000–50,000 patients [9,10]. The primary mechanism in the pathogenesis of MS is an autoimmune process combined with exogenic and environmental factors, genetic predispositions and the patient’s lifestyle [11]. The initial autoimmune response that leads to the development of MS may involve various risk factors. There is compelling evidence that the risk of MS is related to infection with the Epstein-Barr virus, smoking and exposure to tobacco smoke, low vitamin D concentration and obesity during adolescence [12]. A strong association with obesity has also been found in paediatric and youth MS, especially among teenage girls [13].

The results of research on the role of diet during MS treatment are ambiguous, where no consensus has been reached concerning the dietary recommendations for patients with MS [14,15]. In a long-term observation of patients in whom MS occurred before the age of 18 years, each increase of 10% in the consumption of energy provided by fats increased the risk of remission by 56%; while a 10% increase in the consumption of saturated fats more than tripled the risk; and conversely, each additional serving of vegetables in the patient’s diet decreased the risk by 50% [16]. In turn, some meta-analyses have indicated that there is not enough evidence to determine whether supplementation with vitamin D, antioxidants, polyunsaturated fatty acids or other dietary interventions have a significant effect on the health outcomes in patients with MS [17,18,19]. However, the authors of these meta-analyses emphasised that the evidence that has been collected to date remains scarce and is lacking in quality, which makes it difficult to draw clear conclusions. Furthermore, the effect of individual nutrients frequently proves to be too small to prove their direct association with the health indicators, so such analyses may lead to an underestimation of their effect. The results of studies that assessed the diet in a more comprehensive manner showed that a correct diet may improve the health of patients with MS, as well as alleviate some symptoms and support treatment [14,20,21,22,23,24]. Conversely, an incorrect diet may impact the health outcomes, progression of the disease and the patient’s neurological state, primarily through exacerbated inflammation [25]. However, the DPs of patients with MS have not been widely investigated or described in the subject literature across the world [23,26,27], and according to our knowledge, no such research has been conducted in Poland to date.

Many patients with MS display comorbidities dependant on their diet, including obesity, dyslipidaemia, metabolic syndrome, hypertension and cardiovascular diseases (CVDs) [28,29]. These comorbidities are related to an increased activity of MS, in addition to disability, impaired cognitive functions, reduced quality of life, more frequent use of health care and a higher mortality rate [30]. Consequently, the patient’s diet may also modify the course of the disease and exacerbate its symptoms by affecting the occurrence of metabolic disorders. The aim of this study was to analyse the dietary patterns (DPs) of patients with MS and to assess the relationships between these DPs and the observed metabolic disorders.

## 2. Materials and Methods

### 2.1. Data and Sample Collection

The study was conducted between October 2020 and May 2021 among a group of 335 participants affected by MS. Some participants were recruited from among the patients of neurological clinics in the Świętokrzyskie Voivodeship in Poland (*n* = 218). Other participants (*n* = 117) were recruited online from among the patients with MS who were members of the Polish Association for Multiple Sclerosis and from the *NeuroPozytywni* online forum. The inclusion criteria for the study was an age of between 18 and 65 years and a diagnosis of relapsing-remitting, secondary progressive or primary progressive MS as confirmed by a neurologist. The study was conducted with approval from the Bioethics Committee at the Collegium Medicum of the Jan Kochanowski University of Kielce, Poland, issued on 25 April 2020, Approval No.: 24/2020). The collected data were verified and cleaned by removing the respondents who had submitted incomplete personal data and provided unreliable information in the questionnaire. In the end, the basic set of data amounted to 330 patients with MS. The data concerning diet, lifestyle, socioeconomic status, body height and mass, and waist circumference (WC) was collected using the KomPAN^®^ questionnaire [31]. This multicomponent food frequency questionnaire was designed for a Polish population aged from 15–65 years. Research conducted among both healthy and unhealthy subjects has shown KomPAN^®^ to be a reliable tool with acceptable to very good reproducibility [32]. Research has also confirmed that the KomPAN^®^ questionnaire can be recommended for use with data-driven DPs and diet quality scores to describe habitual diets [33].

### 2.2. Demographic and Socioeconomic Data

Demographic data of the respondents included their age, gender, education level (primary school, basic vocational school, secondary school or college), type of employment, region (village, town with <20,000 inhabitants, town with 20,000–100,000 inhabitants or a city with > 100,000 inhabitants), financial situation (modest, comfortable or wealthy) and household situation (very poor, poor, average, good or very good).

### 2.3. Diet

Data were obtained regarding the consumption frequency of 31 food groups. The respondents were asked to choose one of six categories (which were then converted into a daily frequency): never (0 times/day); 1–3 times a month (0.06 times/day); once a week (0.14 times/day); a few times a week (0.5 times/day); once a day (1.0 time/day); or a few times a day (2.0 times/day). Next, in accordance with the questionnaire manual, the overall Diet Quality Index (DQI) was calculated to allow for a combined interpretation of the consumption of food with a potentially beneficial or a potentially harmful effect on health [34]. The respondents also provided information about dieting, supplementation with vitamin D, consumption of meals at consistent times and the number of meals eaten per day (by choosing one of five answers, ranging from one meal a day to five meals or more a day). The nutritional knowledge of the respondents was assessed based on the number of correct answers related to 25 statements about food and diet (true, false or unsure). The level of knowledge was classified as good (17–25 points), satisfactory (9–16 points) or unsatisfactory (0–8 points).

### 2.4. Lifestyle

The following elements of a respondent’s lifestyle were assessed: smoking, physical activity, screen time and sleep time. Past and current smoking habits allowed for ‘yes’ or ‘no’ answers. The respondents chose one of three categories to describe their level of physical activity at work or at school: low—over 70% of sedentary time; moderate—about 50% of sedentary time and 50% active time; and high—about 70% of active time or high-intensity physical labour. Screen time was assessed with the following question: ‘How many hours a day (on average) do you spend watching TV or using a computer (including for work)?’ The respondents chose one of six categories: <2 h/day; 2 to <4 h/day; 4 to <6 h/day; 6 to <8 h/day; 8 to <10 h/day; and ≥10 h/day. The declared sleep time was classified as ≤6 h/day, 7–8 h/day; or ≥9 h/day.

### 2.5. Data Related to Health

The respondents provided information about their body mass and height, which was used to calculate the BMI. A BMI < 18.5 kg/m^2^ was considered to be underweight, 18.5–24.9 kg/m^2^ was considered a normal weight, 25.0–29.9 kg/m^2^ was considered overweight and ≥30 kg/m^2^ was considered to be obese. Data concerning the declared WC was interpreted as abdominal obesity for a WC > 94 cm in men and a WC > 80 cm in women. In addition, the patients with MS were asked to self-assess their health. By using ‘yes’ or ‘no’ answers, they declared the presence of the following metabolic disorders: diagnosed elevated glucose concentration or diabetes, hypertension, elevated triglyceride concentration and decreased HDL-cholesterol concentration. For the patients of the neurological clinics, this information was also confirmed through a medical examination and the appropriate measurements.

### 2.6. Statistical Analyses

The collected data were presented as percentages of the sample for categorical variables and the means (X) and standard deviations (SDs) for continuous variables with a normal distribution. Before the statistical analysis, the normality of the variables was verified with the Kolmogorov-Smirnov test and the Shapiro–Wilk test. Differences between the groups were verified with the Pearson’s chi-squared test for categorical variables and the *t*-test for continuous variables with a normal distribution.

The dietary patterns (DPs) were obtained using a principal component analysis (PCA) with a normalised varimax value. The suitability of the analysis and the number of factors were verified based on the Kaiser–Meyer–Olkin (KMO) coefficient, which amounted to 0.721, and the statistical significance was determined using Bartlett’s test (*p* < 0.0001). A total of 31 variables related to diet were taken into account in the PCA. The number of DPs was identified based on the following criteria: (1) eigenvalues of the correlation of the variables > 1.0; (2) eigenvalue plot; and (3) value of the explained variance. The rotated factor loadings with an absolute value ≥ |0.30| were considered specific for a given DP and were used to label each DP accordingly. The higher the values of the factor loadings, the stronger was the association between a participant’s diet and the DP. The percentage distribution of adiposity and metabolic abnormalities was analysed across the tertiles of the DPs using Pearson’s chi-squared test. A logistic regression verified the associations between the DPs and the adiposity or metabolic outcomes. The odds ratios (ORs) and 95% confidence intervals (95%CIs) were calculated. Two models were created: crude and adjusted for potential confounders, i.e., age (a continuous variable) and gender, smoking and level of education (categorical variables). The modelled variables were abdominal obesity (ref.: absent) and metabolic disorders including hypertension (ref.: not diagnosed), elevated glucose concentration or diabetes (ref.: not diagnosed), reduced HDL-cholesterol concentration (ref.: not diagnosed), elevated triglyceride concentration (ref.: not diagnosed) and at least two metabolic disorders (ref.: ≤ 1 metabolic disorders). With respect to the participants’ adherence to the DPs, the modelled categories were a moderate or high adherence, while the reference category (OR = 1.00) was a low adherence (i.e., the bottom tertile). The statistical analysis was performed using the STATISTICA 13.1 software (StatSoft Poland 2020). A *p*-value < 0.05 was considered to be statistically significant.

## 3. Results

The mean age of the study participants was 41.9 ± 10.8 years. Over three-quarters of the participants were women (76.6%), and almost half lived in large cities (46.1%) (Table 1). Over three-quarters of the participants declared having a comfortable financial situation, and over half declared an average household situation. Most of the participants had completed higher education and were permanently employed. Over 86% of the participants were diagnosed with relapsing-remitting MS (RRMS). This type of MS was more prevalent in women than in men, whereas primary progressive MS (PPMS) was more prevalent in the men (*p* < 0.05). The gender distribution and the prevalence of a given type of MS were consistent with the epidemiological data. Over 56% of the participants assessed their health as being worse due to MS, while only 7.9% assessed it as better. The women declared the occurrence of other chronic diseases more often than the men (25.1% vs. 10.6%). Most of the participants ate three or four meals per day, but only some of them dined at a consistent time. Nearly a quarter of the participants declared following a diet, most frequently, a low-energy diet. Almost three-quarters of the participants used dietary supplements in the last month, with more than half using vitamin D. One-fifth of the participants were current smokers. Over three-quarters slept 7–8 h per day. In addition, over one-quarter declared between 2 and 4 h of screen time; however, a similar percentage declared a total of as much as ≥8 h. The women declared a moderate level of physical activity significantly more often than the men (70.0% vs. 63.6%), whereas the men declared both low and high levels of physical activity more often than the women.

One of the most common metabolic disorders among the participants was abdominal obesity (Table 2), which occurred in as many as 61.3% of the women and 39.0% of the men (*p* < 0.001). Hypertension and an elevated triglyceride concentration were significantly more prevalent in men than in women. Conversely, no significant gender-wise differences were found for abnormal glucose and HDL-cholesterol concentrations. At least two metabolic disorders were diagnosed in 16.1% of the participants, with a similar prevalence between both genders.

The products most often consumed by the participants were vegetables, fruit, refined bread, butter, milk and sweets (Appendix A). However, only 30.9% of the participants ate vegetables a few times per day. The PCA analysis identified three primary DPs, which explained a total of 30% of the variance in the diet (Table 3). The first DP was named ‘Traditional Polish’. It showed a positive correlation with the consumption of cold cuts and sausages, white and red meats, refined bread, sweets, potatoes, cottage cheese, fried foods, butter, cheese, milk and fermented milk beverages. The second DP, named ‘Prudent’, was mainly associated with the consumption of fruit, vegetables, legumes, wholemeal groats, flakes and pasta, fish, eggs, fermented milk beverages, wholegrain bread and juices. The third DP, which was named ‘Fast Food & Convenience Food’, correlated with the consumption of energy and sweetened drinks, fast foods, canned meats, lard, instant soups and alcohol.

The analysis using the DQI confirmed that a high intensity of healthy features was related to a stricter adherence to the Prudent DP and a less strict adherence to the other DPs (Table 4). An inadequate level of nutritional knowledge was associated with a more frequent adherence to the Fast Food & Convenience Food DP.

No significant differences were found in the adherence to each DP depending on the type of MS (Appendix A). However, adherence to the Traditional Polish DP was related to a higher prevalence of overweight and obese conditions, reduced HDL-cholesterol and triglyceride concentration, and at least two metabolic disorders in total (Table 5). No significant differences were observed in the prevalence of metabolic disorders and the nutritional status in relation to adherence to the Prudent DP, whereas adherence to the Food & Convenience Food DP was associated with a higher prevalence of obesity.

The values of the non-adjusted odds ratios showed that the patients who adhered the most to the Traditional Polish DP had a higher likelihood of a low HDL-cholesterol concentration and the occurrence of at least two metabolic disorders in total. Conversely, moderate adherence to the Fast Food & Convenience Food DP was associated with a higher prevalence of a low HDL-cholesterol concentration (Table 6). The analysis of the odds ratios adjusted for age, gender, smoking and level of education showed that adherence to both the Traditional Polish and Fast Food & Convenience Food DPs increased the likelihood of abdominal obesity and a low HDL-cholesterol concentration (Table 7). Adherence to the Prudent DP was not significantly associated with any metabolic disorder.

## 4. Discussion

According to our knowledge, this is the first study that has attempted to identify DPs and analyse their associations with the metabolic risk factors in patients with MS among the Polish population. The Traditional Polish DP identified in this study was primarily characterised by a high consumption of animal products: meat, cold cuts and sausages, butter, milk, cheese, fried foods and a large number of sweets, white bread and potatoes. A higher adherence to this DP by the study participants was related to a higher likelihood of abdominal obesity and a low HDL-cholesterol concentration, and in the non-adjusted model, also with the presence of any two metabolic disorders. The Fast Food & Convenience Food DP was characterised by a high consumption of fast foods, fried foods, canned meats, lard, instant soups and energy and sweetened drinks. It was also related to a higher likelihood of abdominal obesity and an incorrect HDL-cholesterol concentration. Such DPs usually lead to an excessive amount of fat in the diet, especially saturated fatty acids (SFAs), which may affect the expression of genes, lead to the dysbiosis of intestinal microbiota, cause systemic inflammation and contribute to the onset of metabolic disorders and chronic diseases [35,36,37]. Studies conducted by other authors have indicated that a high SFA content in the diet of patients with MS was related to an increased risk, an unfavourable course of the disease [16,34,37] and poor metabolic health [27,37]. In a randomised controlled trial, in which patients with MS followed a low-fat, plant-based diet for 12 months, not only reduced symptoms of fatigue were observed, but also a decrease in BMI (by 0.2 kg/m^2^ per month) and an improvement in the lipid profile [38]. In a prospective study, a disadvantageous lipid profile was associated with a higher degree of disability in patients with MS and resulted in a faster progression of the disease [39]. A different study found that both an incorrect lipid concentration and the presence of disadvantageous genetic polymorphisms caused a faster progression of disability in patients with MS [40]. Relationships were also observed between serum lipid profiles and the MRI outcomes. Higher HDL-cholesterol concentrations were associated with a lower contrast-enhancing lesion volume [41]. Furthermore, a worse outcome on the expanded disability status scale (EDSS) was connected with a higher baseline total and LDL-cholesterol, and with a trend for higher triglyceride concentrations. The mechanisms through which hyperlipidaemia may contribute to MS pathogenesis have not yet been fully explained. According to the research, these mechanisms may involve nuclear receptors and transcription factors that regulate gene expression and inflammatory pathways [42,43]. A study showed that a subtle dysplidaemia occurred during the early stages of MS and correlations were observed between the HDL-cholesterol subfractions and numerous pro-inflammatory cytokines [44].

A high intake of carbohydrates from products such as sweets, white bread and potatoes among the general population leads to abdominal obesity [45,46], triglyceridaemia [45], type 2 diabetes and the complications related to it [47]. In patients with MS, abdominal obesity has been connected with increased disability [48,49]. Currently, little is known about the role of carbohydrates in the diet of patients with MS and their effect on the metabolic risk factors among this group. The subject literature indicates that a patient’s intake of sugar is related to an elevated insulin concentration in the serum, which in turn may lead to inflammation and exacerbate the symptoms of MS [34]. Penesová et al. observed a decreased insulin sensitivity and an upregulation of insulin in response to orally administered glucose in patients with early MS. However, this effect was not connected with chronic inflammation [50]. Conversely, Drehmer et al. concluded that an imbalance between macronutrients in the diet of patients with MS increased the likelihood of abdominal obesity, related to an increased concentration of the pro-inflammatory interleukin IL-6 [51]. Some studies also showed that polysaccharides may have a beneficial effect on the regulation of immunity in MS through an interaction with the intestinal flora [52]. Albrechtsen et al. suggested that a carbohydrate-rich, low-fat diet was associated with an improved physical capacity in patients with MS [53].

Fast food and processed foods are high-calorie products with a low nutrient density and, in addition to a high content of the aforementioned SFAs, they may also contain high amounts of trans fatty acids (TFAs), salt and sugar and a low amount of dietary fibre. A study pointed out that a longer time spent on preparing homemade meals correlated to higher resulting indicators of the diet quality, including a significantly more frequent consumption of fruits, vegetables and salads [54]. In many populations, a frequent consumption of fast food and convenience food is connected with a range of harmful health outcomes, including obesity (especially an increase in visceral fat tissue), disrupted glucose, insulin and leptin homeostasis, lipid and lipoprotein disorders, intestinal dysbiosis, induction of systemic inflammation and oxidative stress, and an increased risk of metabolic syndrome, type 2 diabetes, hypertension, stroke and other cardiovascular diseases [55,56,57]. To date, no studies directly addressing the effect of the consumption of fast food and convenience food on the metabolic health in patients with MS have been published. However, in individuals who consumed fried or fast foods more than once per week, the likelihood of MS was found to be 32.8 times higher compared to those who consumed them less frequently [58]. In a study conducted among adults in Saudi Arabia, eating fast food ≥5 times/week was associated with an increased risk of MS [59]. 

According to the subject literature, more than 75% of consumed sodium comes from processed foods, and an excessive intake of sodium is associated with an increased risk of hypertension [60]. Research has confirmed that prehypertension and hypertension also occur relatively frequently in patients with MS, whereas increased hypertension has been associated with a lowered integrity of white and grey matter of the brain and, consequently, with disability outcomes during the disease progression [61]. Moreover, the Fast Food & Convenience Food DP involved a high intake of alcohol. Research suggests that moderate alcohol consumption may have an anti-inflammatory effect, by increasing the level of interleukin IL-10 and weakening the inflammatory response of monocytes [62]; it may also be protective in relation to metabolic disorders and CVDs [63,64]. However, some analyses have indicated a strongly negative effect of alcohol on health [65]. The available subject literature contained no studies that analysed the relationship between the consumption of alcohol and severity of the metabolic risk factors in patients with MS. Furthermore, no significant differences were found in the severity of MS symptoms between those individuals who had always drank alcohol and the abstainers [66], and a meta-analysis of many studies showed that there is no evidence that the consumption of alcohol is related to an increased risk of MS [67].

In addition, the results of this study showed that the participants who consumed the most fast food and convenience food were the least informed about food and nutrition. Several studies have also confirmed that a better knowledge is associated with healthier nutritional choices, especially with respect to the consumption of fat [68,69].

The third pattern identified in this study (the Prudent DP) involved a high intake of plant products: fruits, vegetables, legumes, vegetable and fruit juices, whole grains, fish, eggs and fermented milk beverages. Adherence to the Prudent DP was not associated with a lower prevalence of metabolic disorders among the participants. Only a slightly lower likelihood of abdominal obesity and hypertension was observed in the participants who adhered the most to the Prudent DP. Studies conducted by other authors have demonstrated that patients with MS who followed the highest-quality diet, i.e., a diet rich in fruits, vegetables, legumes and whole grains, had a lower level of disability and lower indicators of depression [24]. A recent study showed that following the MIND diet, which is rich in vegetables, especially green leafy vegetables and beans, was related to a lower likelihood of MS development [58]. Research indicates that the polyphenols contained in vegetables and beans may reduce the synthesis of pro-inflammatory factors [34]. However, a high intake of the recommended products, such as vegetables, fruits and fish among the participants in this study, with a low consumption of less healthy foods, was apparently insufficient to ensure a beneficial effect on their metabolic health. This may have also been the result of the consumption of low-quality products that contained insufficient amounts of anti-oxidative and anti-inflammatory ingredients. A prospective study conducted recently in the US confirmed that a high content of unhealthy plant products in the diet was associated with a higher risk of overall mortality and CVD-related mortality [70].

### Limitations

The main limitation of this study was its cross-sectional design. Consequently, it was impossible to determine the temporal link between the outcome and the exposure. Furthermore, the data concerning the metabolic risk factors in the patients who participated in the study online was declared, which means that it may have been inaccurate in many cases. In some of the participants, the risk factors may not yet have been diagnosed. The high prevalence of abdominal obesity in this group of participants suggests that the prevalence of other metabolic disorders can be expected to increase over the years, as a result of this abdominal obesity.

## 5. Conclusions

The results of this study have shown that an unhealthy diet is connected with the presence of some metabolic risk factors in patients with MS. The authors of many previous studies have confirmed that the occurrence of comorbidities, including metabolic diseases, may cause a delayed diagnosis and greatly affect the course of MS, degree of disability and the quality of life in patients with MS [40,71], as well as increasing the risk of hospitalisation unrelated to MS, compared to the individuals with no comorbidities [28]. Consequently, appropriate modifications to the diet may lead to an improvement in the metabolic profile of patients with MS and their clinical outcomes.

Because preparing healthy meals is time-consuming, it may constitute a major challenge for patients with MS due to frequent cases of low well-being and limited physical capacity caused by the disease. Consequently, there is also an urgent need to educate patients with MS on the importance of healthy eating, in order to raise their awareness about the effect of food and nutrition on their health and facilitate making appropriate choices in this respect.

## Figures and Tables

**Table 1 nutrients-14-01927-t001:** Characteristics of the sample according to sociodemographic and lifestyle variables.

Variable	Percentage of the Sample (%)			*p*-Value
	Total (*n* = 330)	Men (*n* = 77)	Women (*n* = 253)	
Age (years): mean (SD)	41.9 (10.8)	45.0 (11.3)	41.0 (10.4)	<0.01
Place of residence				
Village	27.3	26.0	27.7	ns
Town < 20 k inhabitants	10.3	10.4	10.3	
Town 20 k–100 k inhabitants	16.4	15.6	16.6	
City > 100 k inhabitants	46.1	48.1	45.5	
Financial situation ^1^				
Modest	10.9	7.8	11.9	ns
Comfortable	74.2	70.1	75.5	
Wealthy	14.8	22.1	12.6	
Household situation				
Very poor	0.6	0.0	0.8	ns
Poor	7.3	2.6	8.8	
Average	53.2	47.4	55.0	
Good	33.9	42.1	31.5	
Very good	4.9	7.9	4.0	
Type of employment				
No, retirement/disability	21.7	26.0	20.4	<0.01
No, unemployed	8.3	1.3	10.4	
Yes, casual work	4.6	2.6	5.2	
Yes, permanent employment	61.5	70.1	58.8	
No, I’m studying	4.0	0.0	5.2	
Education				
Primary school	2.4	3.9	2.0	ns
Basic vocational school	9.2	14.3	7.6	
Secondary school	24.2	29.9	22.4	
College	64.2	51.9	68.0	
Type of MS				
Relapsing-remitting MS (RRMS)	86.1	79.2	88.1	<0.05
Secondary progressive MS (SPMS)	5.8	5.2	5.9	
Primary progressive MS (PPMS)	8.2	15.6	5.9	
Declared health (due to MS)				
Worse	56.1	51.9	57.4	ns
Same	36.0	35.1	36.3	
Better	7.9	13.0	6.4	
Other diagnosed chronic diseases	21.7	10.6	25.1	<0.01
Number of meals per day				
1–2	7.6	11.7	6.3	ns
3	39.4	39.0	39.5	
4	38.8	37.7	39.1	
5 or more	14.2	11.7	15.0	
Eating at consistent times				
Yes	22.4	19.5	23.3	ns
Yes, but only some meals	52.4	55.8	51.4	
No	25.2	24.7	25.3	
Following a diet	24.3	18.2	26.1	ns
Slimming (low-energy)	7.6	10.4	6.7	
Reduced sugar	2.7	3.9	2.4	
Easily digestible, varied	4.8	1.3	5.9	
Vegetarian	3.6	2.6	4.0	
Elimination (gluten- or lactose-free)	3.9	0.0	5.1	
Ketogenic	1.7	0.0	2.0	
Use of vitamin D supplementation	55.3	39.0	57.7	<0.01
Dietary supplements used in the last month	73.9	59.7	78.2	<0.01
Current smoking	19.7	29.9	16.6	<0.01
Smoking in the past	48.9	61.3	45.2	<0.01
Sleep time on working days				
≤6 h/day	22.4	24.7	21.7	ns
7–8 h/day	65.5	63.6	70.0	
≥9 h/day	9.1	11.7	8.3	
Physical activity at work or at school ^2^				
Low	22.4	24.7	21.7	<0.05
Moderate	68.5	63.6	70.0	
High	9.1	11.7	8.3	
Screen time ^3^				
<2 h/day	20.3	24.7	19.0	ns
2 to <4 h/day	25.5	23.4	26.1	
4 to <6 h/day	14.5	14.3	14.6	
6 to <8 h/day	14.2	14.3	14.2	
8 to <10 h/day	16.7	16.9	16.6	
≥10 h/day	8.8	6.5	9.5	

ns—statistically insignificant difference; ^1^ The financial situation was assessed with the following question: ‘How would you describe your household’s overall situation?’; The ‘modest’ category consisted of two answers: ‘We have to be very careful with our daily budget’ and ‘We have enough money for our daily needs, but we need to budget for bigger purchases’; The ‘comfortable’ category consisted of one answer: ‘We have enough money for our needs without particular budgeting’; The ‘wealthy’ category consisted of one answer: ‘We can afford some luxuries’. ^2^ Physical activity at work or at school was categorised as follows: low—over 70% of sedentary time; moderate—about 50% of sedentary time and 50% active time; higher—about 70% of active time or physical labour of a high intensity; ^3^ Screen time was assessed with the following question: ‘How many hours a day (on average) do you spend watching TV or using a computer (including for work)?’.

**Table 2 nutrients-14-01927-t002:** Nutritional status and prevalence of metabolic disorders.

Variable	Percentage of the Sample (%)			*p*-Value
	Total (*n* = 330)	Men (*n* = 77)	Women (*n* = 253)	
BMI (kg/m^2^): mean (SD)	24.5 (4.8)	25.3 (4.4)	24.2 (4.9)	ns
Obesity (BMI ≥ 30.0 kg/m^2^)	12.4	18.2	10.7	ns
WC (cm): mean (SD)	85.1 (13.7)	93.0 (12.6)	82.7 (13.2)	<0.001
Abdominal obesity (WC > 94 cm in men, WC > 80 cm in women) (yes)	56.1	39.0	61.3	<0.001
Diagnosed hypertension (yes)	15.8	23.4	13.4	<0.05
Elevated glucose concentration or diagnosed diabetes (yes)	4.2	6.5	3.6	ns
Low HDL-cholesterol concentration (yes)	7.6	7.8	7.5	ns
Elevated triglyceride concentration (yes)	2.4	5.2	1.6	<0.05
At least 2 metabolic disorders	16.1	16.9	15.8	ns

ns—statistically insignificant difference.

**Table 3 nutrients-14-01927-t003:** Factor-loading matrix for major dietary patterns.

Food Groups	Dietary Patterns (DPs)		
	Factor I Traditional Polish DP	Factor II Prudent DP	Factor III Fast Food & Convenience Food DP
Cold cuts and sausages	0.69		
Refined bread	0.61		
White meats	0.58		
Red meats	0.57		
Sweets	0.51		
Potatoes	0.48		
Cottage cheese	0.48	0.32	
Fried foods	0.47		0.42
Butter	0.46		
Cheese	0.42		
Milk	0.41		
Fermented milk beverages	0.41	0.40	
Legumes	−0.36	0.47	
Fruit		0.60	
Wholemeal groats. flakes and pasta		0.53	
Vegetables		0.51	
Vegetable juices		0.49	
Fish		0.48	
Eggs		0.44	
Wholegrain bread		0.41	
Juices		0.39	
Canned vegetables		0.35	
Energy drinks			0.64
Sweetened drinks			0.61
Fast food			0.61
Canned meats			0.58
Instant soups			0.55
Alcohol			0.47
Lard			0.32
Vegetable oils/margarine			
Refined groats. rice and pasta			
Percentage of variance explained (%)	12.0	9.0	9.0

**Table 4 nutrients-14-01927-t004:** Diet Quality Index (DQI) and nutritional knowledge (%) by adherence to the dietary patterns in the study sample (*n* = 330).

Diet Quality Index (DQI) and Nutritional Knowledge	Total *n* = 330	Traditional Polish			*p*-Value	Prudent			*p*-Value	Fast Food & Convenience Food			*p*-Value
		T1 *n* = 111	T2 *n* = 107	T3 *n* = 112		T1 *n* = 110	T2 *n* = 110	T3 *n* = 110		T1 *n* = 110	T2 *n* = 109	T3 *n* = 111	
DQI													
Low levels of unhealthy traits	1.2	0.0	1.9	1.8	*p* < 0.05	2.7	0.0	0.9	*p* < 0.001	0.0	0.0	3.6	*p* < 0.001
Low intensity of unhealthy features and pro-healthy features	90.0	84.7	89.7	95.5		97.3	96.4	76.4		83.6	91.7	94.6	
High intensity of pro-healthy features	8.8	15.3	8.4	2.7		0.0	3.6	22.7		16.4	8.3	1.8	
Nutritional knowledge level													
Insufficient	12.1	12.6	13.1	10.7	ns	15.5	12.7	8.2	ns	5.5	11.0	19.8	*p* < 0.001
Satisfactory	72.4	73.9	67.3	75.9		76.4	70.0	70.9		74.5	71.6	71.2	
Good	15.5	13.5	19.6	13.4		8.2	17.3	20.9		20.0	17.4	9.0	

T—tertile; ns—statistically insignificant difference; Pearson’s chi-squared test was used to verify differences in sample distribution across the levels of adherence to DP.

**Table 5 nutrients-14-01927-t005:** Occurrence of adiposity and metabolic abnormalities (%) by adherence to the DPs in the study sample (*n* = 330).

Variable	Total *n* = 330	Traditional Polish DP			*p*-Value	Prudent DP			*p*-Value	Fast Food & Convenience Food DP			*p*-Value
		T1 *n* = 111	T2 *n* = 107	T3 *n* = 112		T1 *n* = 110	T2 *n* = 110	T3 *n* = 110		T1 *n* = 110	T2 *n* = 109	T3 *n* = 111	
BMI													
Underweight	5.1	3.6	7.5	4.5	*p* < 0.05	7.3	3.6	4.6	ns	7.3	2.8	5.4	*p* < 0.05
Normal	55.7	66.7	54.2	46.4		51.8	52.7	62.7		58.2	60.6	48.7	
Overweight	26.6	23.4	27.1	29.5		30.0	29.1	20.9		26.4	28.4	25.2	
Obese	12.4	6.3	11.2	19.6		10.9	14.6	11.8		8.2	8.3	20.7	
Abdominal obesity	56.1	52.3	51,4	64.3	ns	58.2	56.4	53.6	ns	50.9	55.0	62.2	ns
Hypertension	15.8	12.6	17.8	17.0	ns	19.1	16.4	11.8	ns	15.5	18.3	13.5	ns
Elevated glucose or diagnosed diabetes	4.2	2.7	4.7	5.4	ns	3.6	5.5	3.6	ns	3.6	5.5	3.6	ns
Low concentrations of HDL-cholesterol	7.6	2.7	6.5	13.4	<0.01	6.4	8.2	8.2	ns	3.6	11.0	8.1	ns
Elevated triglyceride concentrations	2.4	0.9	6.3	2.4	<0.01	1.8	3.6	1.8	ns	2.7	3.7	0.9	ns
At least 2 metabolic disorders	16.1	9.9	15.9	22.3	<0.05	14.5	18.2	15.5	ns	14.5	17.4	16.2	ns

T—tertile; ns—statistically insignificant difference; BMI—body mass index; Pearsonʹs chi-squared test was used to verify differences in sample distribution across the levels of adherence to DP.

**Table 6 nutrients-14-01927-t006:** Crude associations between dietary patterns (DPs) and adiposity and metabolic abnormalities: odds ratios (95% Confidence Intervals (95%CIs).

Dietary Patterns ^1^	Abdominal Obesity	Diagnosed Hypertension	Elevated Glucose or Diagnosed Diabetes	Low Concentrations of HDL-Cholesterol	Elevated Triglyceride Concentrations	At Least 2 Metabolic Disorders
Traditional Polish DP						
Lower—T1	1.00	1.00	1.00	1.00	1.00	1.00
Moderate—T2	0.97 (0.57–1.64)	1.50 (0.71–3.16)	1.76 (0.41–7.57)	2.52 (0.63–10.01)	0.00 (0.00–0.00)	1.72 (0.76–3.86)
Higher—T3	1.64 (0.96–2.81)	1.42 (0.67–2.99)	2.04 (0.50–8.36)	5.57 (1.56–19.81) **	-	2.61 (1.22–5.61) *
Prudent DP						
Lower—T1	1.00	1.00	1.00	1.00	1.00	1.00
Moderate—T2	0.93 (0.54–1.58)	0.83 (0.41–1.66)	1.53 (0.42–5.57)	1.31 (0.47–3.65)	2.04 (0.37–11.36)	1.31 (0.64–2.68)
Higher—T3	0.83 (0.49–1.42)	0.57 (0.27–1.20)	1.00 (0.24–4.10)	1.31 (0.47–3.65)	1.00 (0.14–7.23)	1.07 (0.51–2.25)
Fast food & convenience food DP						
Lower—T1	1.00	1.00	1.00	1.00	1.00	1.00
Moderate—T2	1.18 (0.69–2.00)	1.23 (0.61–2.50)	1.54 (0.42–5.63)	3.28 * (1.02–10.51)	1.36 (0.30–6.22)	1.24 (0.60–2.56)
Higher—T3	1.58 (0.93–2.71)	0.85 (0.40–1.81)	0.99 (0.24–4.06)	2.34 (0.70–7.83)	0.32 (0.03–3.17)	1.15 (0.55–2.36)

^1^ Adherence to the DP is based on subjects’ tertile (T) distribution: bottom tertile = lower adherence (used as the reference level), middle tertile = moderate adherence, upper tertile = higher adherence; Statistical significance (Wald test): * *p* < 0.05, ** *p* < 0.01.

**Table 7 nutrients-14-01927-t007:** Adjusted ^1^ associations between dietary patterns (DPs) and metabolic abnormalities (*n* = 330): odds ratios (95% Confidence Interval).

Dietary Patterns ^2^	Abdominal Obesity	Diagnosed Hypertension	Elevated Glucose Concentrations or Diagnosed Diabetes	Low Concentrations of HDL-Cholesterol	Elevated Triglyceride Concentrations	At Least 2 Metabolic Disorders
Traditional Polish DP						
Lower—T1	1.00	1.00	1.00	1.00	1.00	1.00
Moderate—T2	0.93 (0.54–1.60)	1.04 (0.45–2.41)	1.59 (0.35–7.12)	2.53 (0.63–10.20)	-	1.31 (0.55–3.13)
Higher—T3	1.66 (0.98–2.84) *	0.82 (0.35–1.90)	1.78 (0.41–7.67)	5.22 (1.42–19.12) **	-	1.71 (0.75–3.91)
Prudent DP						
Lower—T1	1.00	1.00	1.00	1.00	1.00	1.00
Moderate—T2	0.86 (0.49–1.51)	0.82 (0.37–1.78)	1.62 (0.42–6.19)	1.38 (0.49–3.88)	3.07 (0.46–20.43)	1.33 (0.61–2.91)
Higher—T3	0.78 (0.44–1.39)	0.92 (0.39–2.13)	1.32 (0.29–5.91)	1.42 (0.47–4.23)	2.97 (0.3–28.95)	1.54 (0.67–3.55)
Fast food & convenience food DP						
Lower—T1	1.00	1.00	1.00	1.00	1.00	1.00
Moderate—T2	1.49 (0.84–2.65)	1.68 (0.74–3.85)	2.17 (0.54–8.76)	3.69 * (1.11–12.32)	2.06 (0.34–12.49)	1.84 (0.80–4.21)
Higher—T3	2.45 (1.33–4.51) **	1.04 (0.42–2.57)	1.52 (0.32–7.35)	2.43 (0.67–8.88)	0.40 (0.03–4.98)	1.64 (0.69–3.90)

^1^ odds ratios adjusted for age (a continuous variable) and gender, smoking, place of residence, financial situation and education (categorical variables); ^2^ Dietary Patterns are based on subjects’ tertile (T) distribution: bottom tertile = lower adherence (used as the reference level), middle tertile = moderate adherence, upper tertile = higher adherence; Statistical significance (Wald test): * *p* < 0.05, ** *p* < 0.01.

## Data Availability

Data presented in this study are available on request from the corresponding author.

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
