# Peer review of "Dietary Patterns and Metabolic Disorders in Polish Adults with Multiple Sclerosis"

_nutrients, 2022, doi:10.3390/nu14091927_

Round 1
Reviewer 1 Report
This is a well-conducted observational study investigating the dietary patterns and prevalence of metabolic disorders among Polish adults with multiple sclerosis. The authors were interested in investigating this due to previous works demonstrating that unhealthy metabolic profile can affect inflammation, progression of disease, and disease severity in multiple sclerosis. This study is conducted appropriately with good sound research design and appropriate reason to conduct the study. However, I have some concern below:
While this study clearly demonstrated a pattern of unhealthy dietary pattern among some proportion of the Polish adults with MS, it is unclear if these dietary pattern would lead to a poorer clinical outcome or prognosis among these patient populations beyond the metabolic syndrome. It would have been a far more interesting study if within the questionnaire, there are inclusions of 'neurological'-phenotype questions. I understand that prior studies had correlated metabolic abnormalities in MS with poorer phenotypes; but I don't understand why the authors did not do this within their specific context of the Polish population.
Throughout the result section, the author tried to explain the differences in some of the measurements between sex. However, it becomes unclear if there are any emerging pattern specific to men/women and there was no conclusion drawn from any of these comparisons. What was the initial motivation to do this comparison to begin-with? If the intent is to observe a dietary pattern driven by sex, this should be more extensively discussed in the discussion and conclusion section. Also, I am concerned if the t-test conducted for these differential analysis had undergone adjustment for multiple-comparisons - the author should clarify on these. I suggest that the authors revisit their description of the Results section to simplify a lot of the description of the statistical findings and comparisons.
Those above are my two major issues with this otherwise great observational study. I understand my concerns above may pertain to study design issues that may not be modifiable at this stage of the study; however these are study design considerations that the author should acknowledge or perhaps consider for future studies.
Otherwise, my minor comments are below regarding word structuring or grammatical errors:
Line 35 - periventricularly?
Line 39 - predominance - do you mean "prevalence"?
Line 40-42 - "The chief role in the ...." is a weird sentence structure. I would advise rewriting and restructuring.
Line 84 - Part of the material (n=117) - author should clarify if these are 117 patients out of the 218 or an additional 117 patients. The total number of patients after cleaning is 330 which I assume means that original total of patients is 218+117 while the way the sentence is written makes it mean that the 117 patient is part of the 'material' of the aforementioned 218 patients.
Results section - primary progressive MS vs relapsing-remitting MS were described without any introduction. Suggest defining these terms either in introduction or beginning of results section.
Line 384 - 386 - "No statistically significant relationships ..." - a confusing sentence. Not sure what the author meant here.
Reviewer 2 Report
The manuscript entitled “Dietary Patterns and Metabolic Disorders in Polish Adults with 2 Multiple Sclerosis” evaluates the effect of the diet on the occurrence of metabolic disorders, showing that an unhealthy diet relates to the presence of some metabolic risk factors in patients with multiple sclerosis. Overall, the manuscript is well written, and the authors have deeply investigated the metabolic risk reporting very well-specified data. Moreover, according to me, the manuscript sends out a very important message to the community, i.e., an urgent need to educate patients with multiple sclerosis on healthy eating, because the appropriate modifications to their diet may improve their metabolic profile and clinical outcomes.
Author Response
April 29, 2022
Nutrients
Reviewer 2
Comments and Suggestions for Authors
The manuscript entitled “Dietary Patterns and Metabolic Disorders in Polish Adults with 2 Multiple Sclerosis” evaluates the effect of the diet on the occurrence of metabolic disorders, showing that an unhealthy diet relates to the presence of some metabolic risk factors in patients with multiple sclerosis. Overall, the manuscript is well written, and the authors have deeply investigated the metabolic risk reporting very well-specified data. Moreover, according to me, the manuscript sends out a very important message to the community, i.e., an urgent need to educate patients with multiple sclerosis on healthy eating, because the appropriate modifications to their diet may improve their metabolic profile and clinical outcomes.
We wish to thank the Reviewer for a positive review of our manuscript.
Sincerely,
Authors of the manuscript